

# Epidemiological scenario of dengue in the state of Manipur during the last 3 years

Leimapokpam Shivadutta Singh[1], Rajkumar Manojkumar Singh[2] and Huidrom Lokhendro Singh[2]

[1] Viral Research and Diagnostic Laboratory, Department of Microbiology, JNIMS, Imphal, Manipur, India
[2] Department of Microbiology, JNIMS, Imphal, Manipur, India

## ABSTRACT

**Background**. The study of disease transmission of dengue fever (DF) is perplexing in the Indian subcontinent as all the four serotypes are circling. Also, there is no efficient epidemiological examination done on dengue cases in Manipur, a north-eastern territory of India.

**Method**. We utilized the dengue information extricated from the lab register of Viral Research and Diagnostic Laboratory (VRDL) from 2016 to 2018. All presumed outpatient and inpatients dengue cases from open and private health-care facilities are incorporated into the VRDL database whose informed consent were gotten.

**Results**. A sum of 1689 instances of associated patients with dengue infection was tried for dengue ELISA test and 272 (16.10%) samples were seen as seropositive. The month-wise conveyance of dengue cases is very intriguing as the three years of study demonstrates a variation design in perception. In all the three years dengue seropositive cases were seen higher in the male populace. Be that as it may, there is no noteworthy incentive to the inspiration of dengue seropositive towards male than female.

**Conclusion**. Our examination exhibits a comparative epidemiological investigation on seroprevalance of dengue in the province of Manipur for three years. This is an endeavour to show epidemiological dengue seroprevalance in the territory of Manipur which in future would be a reference from general wellbeing worries for making up essential move intend to shorten the spread of dengue.

Corresponding author
Leimapokpam Shivadutta Singh,
shivadutta.n@gmail.com

## INTRODUCTION

Dengue is a mosquito borne flavivirus belonging to the family flaviviridae which is the most extensively spread mosquito-borne disease (*Murhekar et al., 2019*). It has five distinct serotypes DENV-1, DENV-2, DENV-3 & DENV-DENV-5 which are distinguished from each other by serological and molecular assays (*Rice, Strauss & Strauss, 1986*; *Borkakoty et al., 2018*). These virus are transferred by female Aedes mosquito especially *Aedes aegypti* and lesser extend *Aedes albopictus* that feed on human blood both indoors and outdoors during dawn to dusk and can be found in tropical and subtropical region particularly dominant in urban environment and spreading out to rural areas (*Mustafa et al., 2015*; *Brady et al., 2014*; *Ahmad et al., 2017*).

In a Chinese medical encyclopaedia in 992 from the Jin Dynasty (265–420 AD), dengue fever was referred as "water poison" associated with flying insects, but the term dengue fever came into general use only after 1828 (*Gupta et al., 2012*). The earliest dengue epidemics occurred almost simultaneously in Asia, Africa, and North America in the 1780s and first clinical case report dates from 1789 of 1780 epidemic in Philadelphia is by Benjamin Rush, who coined the term "break bone fever" because of the symptoms of myalgia and arthralgia (quoted from http://www.globalmedicine.nl/index.php/dengue-fever).

The World Health Organization has revealed dengue as an arboviral malady as one of the eight neglected tropical illnesses (*World Health Organization, 2016*). It is of worldwide general wellbeing concern causing higher dreariness in a large portion of the endemic areas of the world with around 2.5 billion individuals being influenced (*World Health Organization and the Special Programme for Research and Training in Tropical Diseases (TDR), 2009*; *Sodani et al., 2015*). Mostly the urban tenants in tropical and subtropical districts have a higher danger of contracting dengue infection as contrast with other regions (*Halstead, 2007*). According to a WHO report, dengue cases have expanded 30-fold over the last 50 years, and it was evaluated that 96 million instances of dengue happen every year (*WHO, 2009*; *Bhatt et al., 2013*; *Ouyang et al., 2016*). About 75% of current global disease burden due to dengue is borne by southeast Asian region and Western Pacific regions (*Garg et al., 2011*). Falling in the South East Asian area, India has higher incidence of dengue fever leading to threat in health care system (*Kashinkunti & Shiddappa, 2013*). Since its first confirmed report in 1940s dengue infection in India, more and more new states have been reporting the disease in epidemic proportions often inflicting heavy morbidity and mortality (*Kashinkunti & Shiddappa, 2013*).

Early recognition of dengue viral infection disease (DVI) routinely done by the serological test is exceptionally fundamental. IgM antibody is the first immunoglobulin isotype to appear. In a suspected case of dengue, the presence of anti-dengue IgM antibody suggests recent infection. Anti-dengue IgM detection using enzyme-linked immunosorbent assay (ELISA) represents one of the most important advances and has become an invaluable tool for routine dengue diagnosis (*Kuno et al., 1998*; *Hati, 2006*).

The study of disease transmission of dengue fever is intricate and remains inadequately comprehended because of the contribution of status of host, viral and vector which are subject to statistic, financial, conduct and changed cultural components. Various perceptions have raised worries against generally acknowledged epidemiological qualities of dengue (*Guha-Sapir & Schimmer, 2005*; *Guzman et al., 2010*).

Knowledge of local prevalence of infections is critical in guiding clinical work up and treatment. As effective control and preventive programs for dengue infection are based upon improved surveillance data, the objective of this study was to report the seroprevalence of dengue virus infection in Manipur to establish an epidemiological viewpoint in reference to current infection.

## MATERIAL AND METHODS

The present study was conducted at Viral Research and Diagnostic Laboratory (VRDL), Department of Microbiology, JNIMS, Porompat, Imphal East, Manipur during a time of

three (3) years from January 2016 to December 2018. VRDL was set up by Department of Health Research (DHR), Government of India and Indian Council of Medical Research (ICMR) under process of establishing a network of virology diagnostic laboratories in the country with an aim of strengthening laboratory capacity in the country for timely identification of viral diseases and other agents causing significant morbidity.

The state of Manipur is the easternmost state of India, lying between 23°83′N–25°68′N latitude and between 93°03′E–94°78′E longitude, bordering Nagaland in the north, Mizoram in the south, Assam in the west and sharing the international border with Myanmar in the east.

All the samples from patient suspected of having dengue fever (as per WHO guidelines) referred to VRDL from the medical facilities (public or private) of the state and also the samples referred directly by state health authorities for suspected dengue cases were all included for the study. The samples consisted blood samples of both inpatient and outpatient collected during acute phase along with a case report form detailing demographic, clinical, and laboratory characteristics. Serum was separated as soon as possible and refrigerated (2–8 °C) or stored frozen ($\leq-20$ °C), if not tested within 48hrs.Samples obtained within 5 days of onset of fever were qualitatively tested for presence of dengue viral NS1 antigen using the dengue NS1 antigen ELISA (Microlisa J. Mitra & Co. Pvt. Ltd.) supplied by NVBDCP (National Vector Borne Disease Control Program), Manipur where as samples of patients with fever of more than five days duration at time of collection were tested for the presence of anti-dengue IgM antibodies using MAC ELISA NIV (National Institute of Virology), Pune.

We analyzed the laboratory surveillance data and report proportion of laboratory confirmed dengue by time (month and year), place (district and state) and person (age and sex) characteristics. Data were analysed using MS Excel 2007. Types of analysis included proportions and percentage; tests of significance (Chi-square test). $P < 0.05$ was considered statistically significant.

## RESULTS

A total of 1,689 samples of suspected patients of dengue virus infection referred to the VRDL for confirmation of diagnosis of dengue fever over a period of three years, from January 2016 to December 2018 were considered for this study. Out of these, 272 (16.10%) samples were found to positive for dengue virus positive (seropositive) (Table 1). Based on the number of days of fever, 1394 serum samples were tested for anti-dengue IgM antibodies and 295 for NS1antigen.63 (3.73%) samples were serologically positive for NS1 antigen and 209 (12.37%) samples positive for anti-dengue IgM antibodies (Table 1). During this study period, it is seen that dengue was endemically present in the region.

In the year 2016 incidence of dengue began by September and peaked during October and sharply decreased by subsequent months. In the year 2017 the incidence of dengue started by May and abruptly rises till the month of August and declining from the month of September onwards. In 2018 the incidence of dengue were seen sparsely distributed
**Table 1  Year wise distribution of dengue positive cases.** The chi-square statistic is 94.9796. The *p*-value is < 0.00001. The result is significant at *p* < .05.

| Year | Total sample tested | NSI positive | IgM positive | Total |
|---|---|---|---|---|
| 2016 | 251 | 35 (13.94%) | 18 (7.17%) | 53 (21.11%) |
| 2017 | 1,286 | 24 (1.87%) | 181 (14.07%) | 205 (15.94%) |
| 2018 | 152 | 4 (2.63%) | 10 (6.58%) | 14 (9.21%) |
| **Total** | **1,689** | **63 (3.73%)** | **209 (12.37%)** | **272 (16.10%)** |

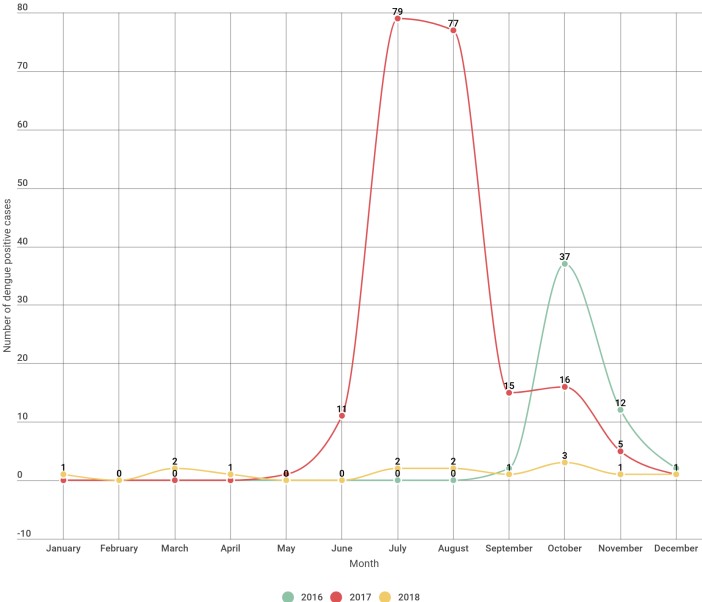

**Figure 1  Dengue positive cases distribution month wise over a three year period.**

throughout the year except in the month of February, May and June with no incidence of dengue (Fig. 1, Table 2).

Overall for the period of three years most of the dengue cases were seen concentrated in the month of June to October (rainy season) and lesser cases in the month of November till May (Table 2).

Seropositive cases in male population were seen little bit higher as compared to that of female during the study period of three years .The proportion of males was found to be higher than females in our study (1.37:1).But such predominance of dengue positivity in male as compared to female is found to be not significant (Table 3).

In 2016 almost all the age group were found to be equally infected by dengue except age group of upto 10 years. For the year 2017 highest positive cases were observed in the age groups of 21–30 followed by upto 10 age groups and least was seen in the case of 41–50 age groups. In the year 2018 the highest positive cases were observed in the age group of

**Table 2  Monthwise/season wise distribution of sero-positive dengue cases.**

| Name of the month | Seropositive (2016) | Seropositive (2017) | Seropositive (2018) | Total (Overall in three years) |
|---|---|---|---|---|
| January | 0 | 0 | 1 | 1 |
| February | 0 | 0 | 0 | 0 |
| March | 0 | 0 | 2 | 2 |
| April | 0 | 0 | 1 | 1 |
| May | 0 | 1 | 0 | 1 |
| June | 0 | 11 | 0 | 11 |
| July | 0 | 79 | 2 | 81 |
| August | 0 | 77 | 2 | 79 |
| September | 2 | 15 | 1 | 18 |
| October | 37 | 16 | 3 | 56 |
| November | 12 | 5 | 1 | 18 |
| December | 2 | 1 | 1 | 4 |
| **Total** | **53** | **205** | **14** | **272** |

**Table 3  Gender wise dengue sero-positive distribution.** The chi-square statistic is 2.1314. The $p$-value is .344481. The result is *not* significant at $p < .05$.

| Year | Positive males | Positive females | Total sero-positive sample |
|---|---|---|---|
| 2016 | 27 | 26 | 53 |
| 2017 | 120 | 85 | 205 |
| 2018 | 10 | 4 | 14 |

**Table 4  Age wise sero-positive cases.**

| Age | Positive 2016 | Positive 2017 | Positive 2018 | Total |
|---|---|---|---|---|
| Upto 10 | 3 | 47 | 0 | 50 (18.38%) |
| 11–20 | 11 | 27 | 6 | 64 (23.53%) |
| 21–30 | 11 | 54 | 4 | 69 (25.37%) |
| 31–40 | 11 | 22 | 1 | 34 (12.5%) |
| 41–50 | 10 | 15 | 1 | 26 (9.56%) |
| More than 50 | 7 | 20 | 2 | 29 (10.66%) |
| | | | | 272 |

11–20 years followed by 21–30 age groups while least was seen in the age group of 31–40 and 41–50. (Table 4 & Fig. 2).

The dengue positive cases were seen distributed in 6 districts in the year 2016. Imphal West district showing the highest positivity followed by Imphal East. In the year 2017 distribution of dengue positive cases were seen in all the districts of Manipur with Churachandpur district having highest positivity followed by Imphal East, Imphal West. While the least positive cases were observed in Tamenglong district. In the case of 2018 the highest positive cases were from Imphal East district and Thoubal district. While three

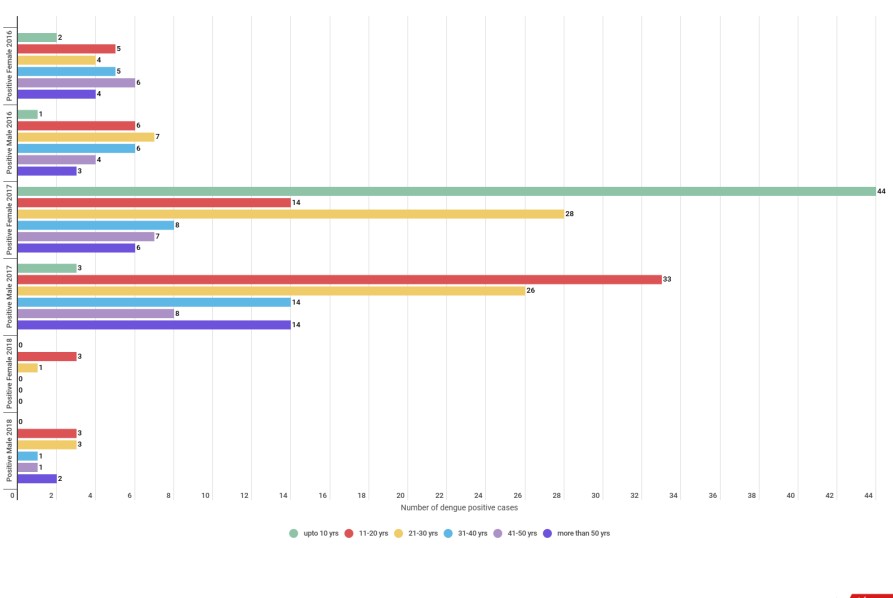

**Figure 2  Gender wise dengue positive cases among different age groups.**

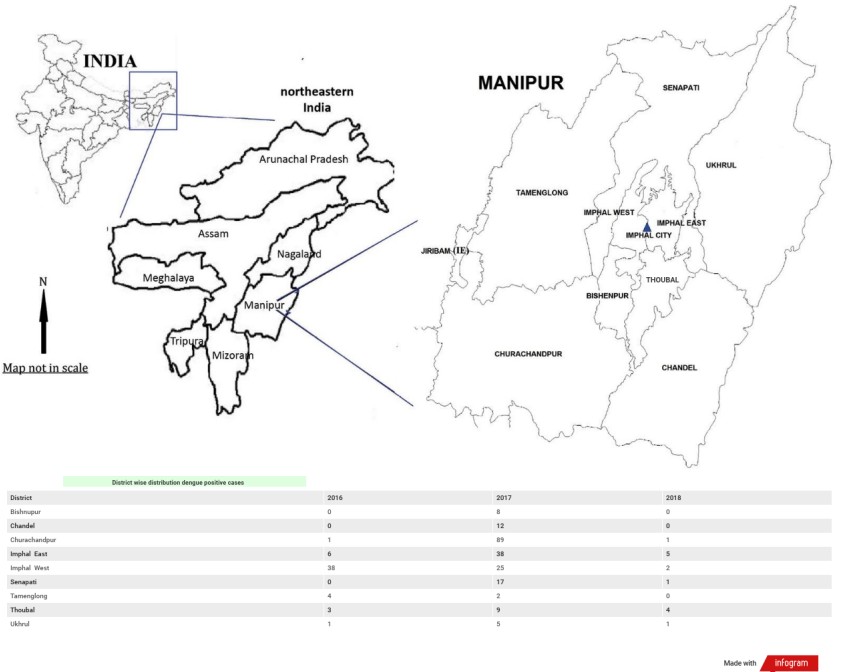

| District | 2016 | 2017 | 2018 |
|---|---|---|---|
| Bishnupur | 0 | 8 | 0 |
| Chandel | 0 | 12 | 0 |
| Churachandpur | 1 | 89 | 1 |
| Imphal East | 6 | 38 | 5 |
| Imphal West | 38 | 25 | 2 |
| Senapati | 0 | 17 | 1 |
| Tamenglong | 4 | 2 | 0 |
| Thoubal | 3 | 9 | 4 |
| Ukhrul | 1 | 5 | 1 |

**Figure 3  District wise distribution of dengue positive cases in Manipur: 2016–2018.**

district namely Bishnupur, Chandel and Tamenglong did not have any positive cases (Fig. 3).

Overall in the three years maximum dengue positive cases were seen concentrated in Churachandpur district followed by Imphal West, Imphal East, Senapati, Thoubal,

Chandel. While Bishnupur, Ukhrul and Tamenglong districts had least dengue positive cases concentration.

## DISCUSSION

Spread of awareness of dengue infection among health care workers and public has paved the way of increased serological tests leading to higher rate of detection of dengue cases over the past few years (*Sodani et al., 2015*). The endemicity of dengue is spreading and has witnessed a 30-fold increase with rapid expansion to more than 100 countries in Africa, America, Eastern Mediterranean, South-East Asia and Western Pacific areas from urban to rural settings and worst affected regions are South-East Asia and Western pacific regions (*Singla et al., 2016*; *Kumar et al., 2017*). In this study 16.10% cases were dengue positive serologically which is lower than the findings of others (*Singla et al., 2016*; *Garg et al., 2011*; *Chitkara et al., 2018*). But the positivity rate of study is found to higher as compared to report of other studies (*Sherchand et al., 2001*; *Shah et al., 2012*; *Bin Yunus et al., 2002*). Such variation in seropositive rate could be due to different geographical areas and climatic conditions (*Kashinkunti & Shiddappa, 2013*).

In India, the vulnerability of dengue has increased in recent years due to rapid urbanization, lifestyle changes and deficient water management including improper water storage practices in urban, peri-urban and rural areas, leading to proliferation of mosquito breeding sites (*Kumar et al., 2017*).

Determining the differences infection rate among male and female is important for public health control programmes. In this study higher incidence of dengue infection among male as compared to that of female was seen. Such higher incidence of dengue infection among male population than female population was similarly reported in other studies (*Antony & Celine, 2014*; *Garg et al., 2011*; *Kashinkunti & Shiddappa, 2013*) and could be due to extensive exposure of males to dengue-carrying mosquitoes or differences in the healthcare-seeking behaviour of males and females (*Anker & Arima, 2011*; *Arima, Chiew & Matsui, 2015*).

Dengue infection was found in all the age groups in our study but highest was seen in the age group of 21–30 yrs which is in accordance with the findings of the study done by *Sodani et al. (2015)*, *Paul et al. (2018)* and *Gupta et al. (2016)*. Dengue infection is not age specific and not only children but adults are also equally under threat of dengue infection.

To identify the seasonal variation of the dengue infection, analysis of the data on monthly basis were done. A gradual increase in dengue positivity was noticed from September with a peak in October, in the year 2016 which is quite close to finding by *Garg et al. (2011)*. But in the year 2017 the dengue cases started to increase from the month of May with a peak in July, August. However, the seasonal variation in the year 2018 seems quite different with low level of dengue cases and uneven distribution pattern throughout the year. Such uneven pattern of seasonal variation of dengue infection is quite different from the studies done by *Kashinkunti & Shiddappa (2013)* and *Garg et al. (2011)*. Such pattern is an indication for weak relationship between monthly mean temperature and incidence of dengue as indicated by studies done by *Hay et al. (2002)*. As revealed by the study of *Guha-Sapir & Schimmer (2005)* the present study supports the oversimplification of the relationship

between temperature, rainfall and increasing vector-borne disease. However the indication of overall dengue infection in the three years of study seen mostly during rainy season of the state (June–October) indicates its correlation with monsoon season. Moreover anthropogenic climate change due to human activities such as extensive urbanization, explosive growth of population, deforestation/degradation of forests for industrialization, increasing emissions of fossil fuel, waste disposal etc. may have paved a way for increase of vector borne diseases such as dengue (*Sethi & Bidyarani Sharma, 2017*).

## CONCLUSION

The present study reveals that the prevalence of dengue cases in the State of Manipur with differential pattern of distribution with respect to geographical, age wise and season wise. The findings in the present study extend the knowledge of the geographical distribution and seroprevalance of dengue in the state of Manipur for the last three years. This study is the first to provide a consistently derived overview of dengue seropositivity data for the state. Given that the majority of dengue infections are clinically asymptomatic, and that the disease is greatly underreported, these results provide distinctive information on dengue transmission per age group in the different districts of the state, and will be invaluable in future modeling studies that explore the temporal and spatial distribution of dengue infection.

This is an attempt to present epidemiological dengue seroprevalance in the state of Manipur which in future would be a reference from public health concerns for taking up necessary action plan to curtail the spread of dengue. Surveillance of dengue cases is still warranted to be vigilant about any new genotype introduction in the endemic districts.

## ACKNOWLEDGEMENTS

We acknowledge our laboratory technician Mr Kh Bipin Singh and Mrs Sulla Yumnam for their technical support.

### Funding

This study is supported by Department of Health Research-Indian Council of Medical Research, Ministry of Health and Family Welfare, Govt. of India. The funders had no role in study design, data collection and analysis, decision to publish, or preparation of the manuscript.

### Grant Disclosures

The following grant information was disclosed by the authors:
Department of Health Research-Indian Council of Medical Research.
Ministry of Health and Family Welfare, Govt. of India.

### Competing Interests

The authors declare there are no competing interests.

## Author Contributions

- Leimapokpam Shivadutta Singh conceived and designed the experiments, performed the experiments, analyzed the data, prepared figures and/or tables, authored or reviewed drafts of the paper, and approved the final draft.
- Rajkumar Manojkumar Singh and Huidrom Lokhendro Singh analyzed the data, authored or reviewed drafts of the paper, and approved the final draft.

## Ethics

The following information was supplied relating to ethical approvals (i.e., approving body and any reference numbers):

The office of the Institutional Ethics Committee, Jawaharlal Nehru Institute of Medical Sciences, Imphal deemed the research exempt from review as the research is based on secondary data (Proposal No. 168/08/2019).

## Data Availability

Data is available as a Supplemental File.

## Supplemental Information

Supplemental information for this article can be found online at http://dx.doi.org/10.7717/peerj.8518#supplemental-information.

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
