# Peer review of "Epidemiological scenario of dengue in the state of Manipur during the last 3 years"

_PeerJ, doi:10.7717/peerj.8518_

## Round 0.1 · original submission · Major Revisions

Based on the reviews of two external experts, I would like you to revise and resubmit your paper. Please note, the work may be publishable after addressing the comments of two reviewers. Both point out the need to better address the biology of dengue and further use of the literature in a review. Please use these reports a guide to revise your paper. Thank you.

Reviewer 1 ·

Basic reporting

the paper is clear and unambiguous, however still some English errors both spelling and gramma to be improved.

Experimental design

Informed consent was mention when including patients which is not necessary if anonymous. those activities may deducted some important information of those who have better knowledge.
Research question is clear
Method clear

Validity of the findings

Still some issues relate to the representative of data. the refused rate not been reported.
Some redundancy reporting data
I have no idea about the test they compare gender from line 100-106. p value could be enough to report result.
Most of the cases found in 2017 as well as the change in peaking time, it need to have clearer explanation for that time.
In year 2017, most of positive case is confirmed by IgM while NSI rather lower than 2016 and 2018. Any suggestion for that change? any change in method or reagent of test?

Additional comments

Abstract too long compare to the body text. Conclusion also need to be compact and show the main finding for this research. When explain about the epidemic, it is needed to compare with a long period of data, usually, for dengue is 10 years of data to see the seasonality and pattern. Other information about case definition, test used also need to be clarified.

Reviewer 2 ·

Basic reporting

English language corrections are required throughout the manuscript.

Enough literature survey has not done by the authors.

Repeated representation of data in the form of table and figures.

Article structure is needed to be correct.

Experimental design

Scope of the article is good but lack of proper representation.

Methods: Should provide the information about what kind of statistical analysis has been done for the study and how you have done and which software were used is mandatory.
No description of the study area and study area map.

Validity of the findings

Results:
Table-1 and figure-1 represent the same information hence one should be deleted.
Lines 79-81, 84-85 and 86-87 give the same information and the sentences are repeated too many times.
Line-90: The author did not show any temperature data but compared cases are declining with temperature. Should not state such statements without proper analysis.
Again Table-2 & Figure-2 describes the same information one should delete either table or figure.
All figures and tables describe the same information hence keep either tables only or figures only.
Instead of describing the temporal distribution of dengue by district wise, should have shown spatially for easy understanding.
Is there any specific reason that males are more prone to dengue than females.
Instead of comparing with other authors why this seasonal difference of dengue was observed among the years should discuss briefly. Similarly, why particular the age group is prone for dengue should discuss.
148-150: Need reference for the statement.

Additional comments

Abstract: Line 12-15 is not required in the abstract.
The result is not significant at p < .05.: May be removed from abstract.

Introduction
1sr paragraph: Space after the full stop.

General: English language corrections are required throughout the manuscript.

Reviewer 3 ·

Basic reporting

See below

Experimental design

See below

Validity of the findings

See below

Additional comments

The manuscript by Singh et al describes the epidemiology of dengue from one Viral Research and Diagnostic Laboratory from Imphal East. I have the following comments on the manuscript

Introduction
1. Authors mention that they conducted a study to focussing various epidemiological factors and also for the seroprevalence pattern of dengue virus infection over the last 3 years in Manipur. Have you conducted a sero-prevalence, by testing for IgG antibodies against dengue?
2. It would be good if you could clearly spell the objective of your analysis

Methods
3. For the sake of readers, would you elaborate more about the Viral Research and Diagnostic Laboratory Network? Which patients are included in the surveillance? Whether hospitalized or also those attending OPD? What is the case definition followed? Whether all suspected patients are tested or only a sample of patients? What proportion of total suspected patients were in fact tested in each of the three years? Do you have any follow-up data from the patients? Do you have information about severity as well as case fatality of dengue?
4. You mention that sera are tested for IgM antibodies, NS1 antibodies? Can you please clarify if all sera are tested for both NS1 and IgM? Which kit is used for diagnosis? What is the sensitivity and specificity of these tests?
5. You mentioned that temporal patterns of dengue cases was explored by plotting monthly incidence for the study period. JNIMS is the tertiary care hospital in Imphal. How correct is your statement that what you calculated is incidence, when in fact all patients in Imphal district (including those attending private sector) are not included in the numerator?

Results
6. You mention that 1689 suspected patients of dengue (please provide the definition) were referred to VRDL. How many suspected patients of dengue in fact attending during this period?
7. Fig 1 – Why the number of cases investigated during 2017 was more, as compared to 2016 and 2018? How can you explain such a rise during 2017?
8. Fig 1 – provide Y axis label
9. Authors attribute the decrease in the sero-positivity over 2016-18 to the public health issues caused by spread of dengue in Manipur. What exactly do you mean by PH issues? Can you elaborate this point?
10. I am not sure if you have provided adequate data to support monthly increase/decrease in number of cases by temperate. No analysis has been done to show if the two variables are correlated
11. Please calculate dengue positivity (#Positive/#Tested) in males and females and then compare the proportion. The way it is presented in the manuscript, is less meaningful.
12. It appears that the virology lab received samples from six neighbouring districts. If the patients/samples from neighbouring districts are included in the analysis, how appropriate is your calculation of incidence, as well as seasonal trend of disease? How sure are you that the referral pattern from different districts did not change over three years?
13. Did you test 1417 sera which were negative for dengue, for other aetiologies?
14. What were clinical differences among dengue sero-positive and negative cases?

Discussion
15. First paragraph - general statement. Suggest to provide reference
16. Studies by Atul, Jimmy, Mohan et al? Where were these studies conducted? Are your methods comparable to their studies?
17. Too superficial.

---

## Round 0.2 · accepted · Accept

Thank you for carefully revising your draft and meeting the comments of the reviewers. This paper will make a nice addition to the dengue literature.